psychology

word learning, actions, consistency, variability, cross-domain influences

**Author for correspondence:**
Sarah F. V. Eiteljoerge
e-mail: sarah.eiteljoerge@psych.uni-goettingen.de

# Consistency of co-occurring actions influences young children's word learning

Sarah F. V. Eiteljoerge[1,2], Maurits Adam[3], Birgit Elsner[3] and Nivedita Mani[1,2]

[1]Department for Psychology of Language, University of Goettingen, Goettingen, Germany
[2]Leibniz ScienceCampus 'Primate Cognition', Goettingen, Germany
[3]Developmental Psychology, Department of Psychology, University of Potsdam, Potsdam, Germany

 SFVE, 0000-0001-6141-7924

Communication with young children is often multimodal in nature, involving, for example, language and actions. The simultaneous presentation of information from both domains may boost language learning by highlighting the connection between an object and a word, owing to temporal overlap in the presentation of multimodal input. However, the overlap is not merely temporal but can also covary in the extent to which particular actions co-occur with particular words and objects, e.g. carers typically produce a hopping action when talking about rabbits and a snapping action for crocodiles. The frequency with which actions and words co-occurs in the presence of the referents of these words may also impact young children's word learning. We, therefore, examined the extent to which consistency in the co-occurrence of particular actions and words impacted children's learning of novel word–object associations. Children (18 months, 30 months and 36–48 months) and adults were presented with two novel objects and heard their novel labels while different actions were performed on these objects, such that the particular actions and word–object pairings always co-occurred (*Consistent* group) or varied across trials (*Inconsistent* group). At test, participants saw both objects and heard one of the labels to examine whether participants recognized the target object upon hearing its label. Growth curve models revealed that 18-month-olds did not learn words for objects in either condition, and 30-month-old and 36- to 48-month-old children learned words for objects only in the Consistent condition, in contrast to adults who learned words for objects independent of the actions presented. Thus, consistency in the multimodal input influenced word learning in early childhood but not in adulthood. In terms of a dynamic systems account of word learning, our study shows how multimodal learning settings interact with the child's perceptual abilities to shape the learning experience.

# 1. Introduction

Infants grow up in a multimodal world, where language input is typically embedded in a contextual framework. For example, this input can be provided with concurrently presented gestures like pointing and actions like the hopping of a rabbit. Indeed, words are often accompanied by actions [1], and actions are often accompanied by words (e.g. [2,3]). In a study by Gogate *et al.* [4], temporal alignment of parental language and actions in naming events was around 76% for preverbal infants, underscoring the co-occurrence of speech and action in early communication with infants. This rich environment provides infants with the opportunity to experience and learn from different sources and senses but also requires the learner to play an active part in shaping her learning experience, filtering out what is relevant for her. How does a child determine what to focus on when confronted with simultaneous speech and action? Does information from one domain interfere with the processing in the other domain, or does cross-domain information foster processing on another domain? We examined these questions in the current study by investigating the extent to which consistency in the co-occurrence of particular actions and word–object pairings influenced children's learning of the novel word–object associations: in a word learning task, two groups of 18-, 30-month-old and 3- to 4-year-old children and adults saw two objects and heard their labels while the objects moved in either the same (*Consistent* group) or in a varied (*Inconsistent* group) manner across successive presentations of the word–object association. At test, participants' learning of the word–object associations was tested in a preferential looking paradigm [5] to see whether action consistency had an influence on their word learning.

The literature on early speech perception and word learning documents the impressive pace and flexibility with which infants acquire their native language, typically understanding and producing their first words even before their first birthday (e.g. [6,7]). Similarly, infants display rapid development in the action domain from early on. By three months, infants can represent goal-based actions [8], and at least by nine months they seem to understand the implications of actions [9]. Thus, already at 1 year of age, infants show impressive language and action processing skills.

Furthermore, words and actions often co-occur in the input: as mentioned above, Gogate *et al.* found in their study that temporal alignment of actions accompanying carer speech was around 76% during year 1 [4], and 2-year-old children accompany their own speech with actions [10]. Much work by Gogate and colleagues has investigated the role of mothers' action presentations during semi-structured free play and finds that mothers favour certain action types in temporal synchrony with language in communicative interactions with their six- to eight-month-olds [11]. With time, the temporal synchrony of actions and words in maternal communication with infants reduces, adapting to the child's developmental needs and their reduced reliance on synchronicity in multimodal interactions [4].

The high co-occurrence of words and actions in the input of the child might be accompanied by cross-domain influences on word and action learning. Indeed, studies have shown that young infants seem to benefit from multimodal input when learning novel words. In these studies, the concurrent presentation of actions and words or syllables helped six- to eight-month-olds to learn the word–object mappings, as long as actions and words were temporally synchronous and infants attended to the presentation of the multimodal input (e.g. [12]). Similarly, in experimentally controlled habituation paradigms, eight-month-olds were better able to learn word–object mappings when the objects were presented with temporally synchronous actions, with some actions being favoured compared to others (e.g. *shaking* compared to *sideways* [1]). Indeed, Werker *et al.* [13] report that 14-month-olds learned the associations between words and their referents *only* when these objects were moving, but not when they were stationary. These results suggest that concurrently presented actions foster word learning in young children, potentially due to such multimodal content highlighting the connection between the word and the object. In contrast, Puccini & Liszkowski [14] find that 15-month-olds learned words for objects, but not when they were accompanied by a referential gesture (e.g. a fist moving up and down). Like the study by Matatyaho-Bullaro *et al.* [1], these results suggest that the nature of the concurrently presented actions or, in this case, gestures, may influence young children's word learning.

Likewise, language input can influence children's interpretation of action demonstrations. For example, language can help structure action sequences (i.e. acoustic packaging [15,16]) and modulate children's representation of goal-directed actions [2] by highlighting the relevance of an action and guiding children's imitation of these actions (see also [3]). Language can also facilitate the comparison of actions in infants as early as 10 months of age and help infants to understand actions as being goal-directed [17].

Thus, the concurrent presentation of information from the word and action domains fosters learning in both domains. The studies reviewed above have all, however, examined the temporal synchrony between (particular) action demonstrations and language input, i.e. the effect of actions being presented concurrently with language. Here, we extend this to examine the influence of the consistency of concurrently presented action demonstrations on word learning. By consistency, we refer to the consistency with which certain actions accompany certain word–object associations. For example, when parents introduce children to new word–object associations, parents might make the rabbit hop or wiggle its ears. Thus, these action–word–object triads can either be consistent, namely that the same action accompanies successive presentations of a word–object association (i.e. the rabbit is always hopping), or inconsistent in that different actions accompany each presentation of a word–object association (i.e. the rabbit is sometimes hopping and sometimes wiggling his ears).

Indeed, consistency in the input to children appears to have a strong influence on early word learning. Consistent mappings of words and objects are obviously a prerequisite for word learning. This is best observed in cross-situational learning paradigms where learning is based on 12- to 14-month-olds, tracking the frequency with which distinct words are presented with distinct objects across successive presentations (e.g. [18]). Moreover, studies have shown that consistency in referent location [19,20] and consistent spatial, temporal and linguistic contexts [21] support word learning in early childhood. Also, studies on repetitive storytelling find that 3-year-olds learned novel words better when being read the same book (containing the novel words) relative to being read different books with the same novel words [22,23]. Finally, demonstrating the benefit of syntactic consistency, Schwab & Lew-Williams [24] show that consistent sentence structure similarly helps 2-year-olds learn new words. Taken together, consistency plays a crucial role in the acquisition of early word–object mappings, impacting the strength with which such mappings are both formed and later accessed.

However, if more than two aspects in this context co-occur, the intended referents of the label may be ambiguous: for instance, if a particular action demonstration consistently accompanies presentations of a word–object association, the word could either refer to the action or to the object due to their similarly high rates of co-occurrence. If, however, the actions vary while the word and object are consistently presented together, the variability in the action presentation might help in disambiguating the object as the intended referent of the word.

Variability has typically been shown to play a role in structuring abstract knowledge, for example, in categorization tasks. For example, variability in visual appearance across trials facilitated learning of perceptual categorization in infants as young as six to seven months [25]. Also, 10-month-olds learned to discriminate between a typical and an atypical member of a category, only when the objects in that category varied strongly in their perceptual appearance, but not when they were perceptually similar [26]. Furthermore, hearing the same words produced by different speakers helped 14-month-olds to discriminate between the previously heard word and a similar sounding word, suggesting that variability in the phonetic detail of speakers may help a child to grasp subtle differences between those words [27,28].

Twomey *et al.* [29] suggest, however, that there are limits to the effects of variability on learning: although 30-month-olds learned labels for categories when the objects varied in colour (but not when they were in identical colours), children did not learn category labels when objects varied in shape *and* colour. Thus, too much variability disrupted children's learning of category labels. In contrast, Junge *et al.* [30] presented nine-month-olds with either constant picture–word pairings (i.e. always the same picture of a cat when hearing the label *cat*), or varying pictures of the same object (i.e. different cats when hearing *cat*). Both groups showed learning, indicating that neither consistency nor variability influenced their word recognition. Furthermore, Twomey *et al.* [31] tested 2-year-olds' word learning abilities while the background colours of the screen either varied in colour or stayed consistently white. They found that children only showed target recognition in the variable condition which led them to assume that decontextualization helped the children to form strong word representations. Nonetheless, they also suggest that increased variability might disrupt successful learning and that this might relate to the learning environment: reduced variability might help in rich learning environments and increased background variability may boost learning in simpler learning environments (see [29,32]). Thus, the effect of variability may vary across contexts and guide attention allocation in different ways, thereby influencing learning behaviour (see [33]).

Importantly, as the literature reviewed above suggests, both consistency and variability in the input are required for learning because they seem to serve different functions: consistency can help to form strong representations and to deepen your knowledge. In contrast, variability can help to categorize

these representations and to broaden your knowledge. Accordingly, both aspects in combination can lead to consolidated and diverse knowledge of the world [34,35].

In the present study, we ask whether young children and adults better learn the labels of novel objects when particular actions *consistently* co-occur with the novel word–object pairings relative to when they are presented with *inconsistent* actions, i.e. when distinct word–object mappings are accompanied by varying action demonstrations.

We tested three groups of children and adults as a control group. We chose 18-month-olds and 30-month-olds to capture children on either side of the vocabulary spurt. For the 18-month-olds, we expected that they would learn words for objects, but that their learning would be influenced by the consistency of actions. For the 30-month-olds, being past the vocabulary spurt, we expected that the consistency of actions would not influence their word learning, and that they would thus learn in both conditions. 3- to 4-year-olds were added on later following the results of the 30-month-olds, and we expected them to learn in both conditions as they are experienced word learners and might not be as easily distracted by actions. Therefore, this age group might be most similar to the adults whom we expected to learn in both conditions.

## 2. Methods

### 2.1. Participants

Fifty-four German monolingual 18-month-olds (range = 17.23–22.55 months; mean = 18.07, 25 girls) and sixty-five 30-month-olds (range = 27.09–32.12 months; mean = 29.72, 30 girls) and fifty-nine 3- to 4-year-olds (range = 38.89–47.9 months; mean = 43.44, 27 girls) participated in the experiment. In addition, 60 adults were tested as a control group (range = 19−50 years; mean = 26.32, 38 women). Twenty-seven additional participants were tested but excluded from the analysis because of unwillingness to participate (2), technical failure (2), calibration issues (4), bilingualism (2), preterm birth (2), impaired hearing (2), familiarity with the presented novel object (2), or insufficient data (11, see Preprocessing for details). Children were recruited from the group's database, and participation was rewarded with a book. In order to control for children's language abilities, carers of 18- and 30-month-olds completed the German adaption of the FRAKIS (Fragebogen zur frühkindlichen Sprachentwicklung [36]). To test the language abilities of 3- to 4-year-olds we used the SETK3-5 (Sprachentwicklungstest für drei- bis fünfjährige Kinder [37]). Adults were mostly students of the University and were rewarded with either 0.5 course credit points or 4 €. Ethics approval was granted by the University of Göttingen (Project 123).

### 2.2. Stimuli

We selected two pseudo-words in keeping with German phonotactic constraints (Tanu and Löki), two arbitrary actions (explained in further detail below), and two novel objects (a yellow and a blue soft toy from https://www.giantmicrobes.com/us/, figure 1). The auditory stimuli were recorded by a female German native speaker in infant-directed speech. The labels were embedded in a carrier phrase in both the training phase (e.g. 'Schau mal, ein Tanu!','Look, a Tanu!') and test phase (e.g. 'Wo ist denn das Tanu?', 'Where is the Tanu?').

The video stimuli consisted of training and test videos. In the training videos, participants saw a hand (with the arm of the agent being visible) moving the objects according to the two selected actions starting from the middle of the screen. We recorded separate videos of both actions being performed on both objects, such that across participants, videos of each distinct action being performed on each object were presented the same number of times. The first action involved a hand moving the object upwards while moving the object side-to-side (from left to right) with each increment in height. It started in the lower middle of the screen, went up and back down again. The second action involved a hand moving the object sideways, while tilting back and forth, moving first to one side of the screen, then back to the other, and ending up in the middle of the screen again. Both actions filled the whole 7 s of the trial presentation. The auditory stimuli, i.e. the labels for the objects embedded in carrier phrases, were presented at the same time to ensure temporal synchronicity of actions and language [1]. Each video in the training phase was 7 s long and $720 \times 420$ pixels in size.

Across the test trials, infants saw the two objects (still images) side-by-side on screen for 5 s and were led to fixate the target using carrier phrases (e.g. 'Where is the Tanu?') such that the target label began 2.5 s into the video. Individual images of the two objects in the test phase each $640 \times 480$ pixels, and areas of interest for the analysis were defined accordingly.

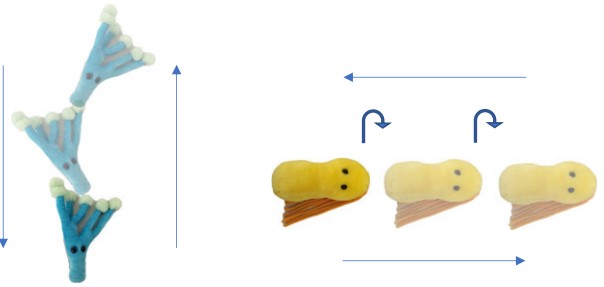

**Figure 1.** Blue and yellow toys were used as novel objects. As novel actions, an upward movement with leaning to the sides and a sideways movement with tilting backwards and forwards were used.

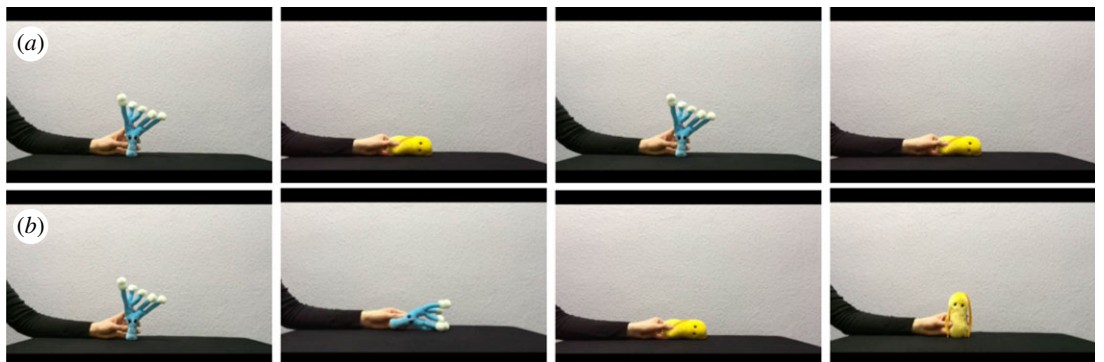

**Figure 2.** Example of the training phase. In panel (*a*), each object is associated with one label and one action (Consistent condition). In panel (*b*), each object is associated with one label and both actions (Inconsistent condition). Each video in the training phase lasted 7 s, and an attention getter was presented before each trial. All videos are available on OSF under osf.io/tndj7.

## 2.3. Procedure

Participants sat in a dimly lit and quiet experimental room at a distance of 65 cm from a TV screen (92 × 50 cm). Children sat either in a car seat or on the parent's lap. A remote eye tracker (Tobii X 120), set on a platform underneath the TV screen, was used to record gaze data at 60 Hz. The software E-Prime was used to present the stimuli. Auditory stimuli were presented via two loudspeakers situated above the television screen. Two video cameras centred above the screen served to monitor the participant during the experiment. Calibration was conducted in Tobii Studio using a 5-point calibration procedure, in which a red point appeared in all four corners and the centre of the screen. The experiment began following successful calibration. Each trial started with a Teletubby serving as a fixation cross in the middle of the screen against a black background, followed by the stimulus presentation. The videos in the training phase were presented in the centre of the screen. Pictures in the test phase appeared next to each other in the centre of the left and right half of the screen.

## 2.4. Experimental design

Each participant was presented with a yoked training and a test phase in a preferential looking paradigm [5]. The training phase consisted of four trials, with each trial presenting the participant with the individual objects in motion accompanied by the label for this object (figure 2). Participants were presented with two trials for each word–object association, and the pairing of objects with labels was counterbalanced across participants. Thus, *Tanu* was the blue object for half of the participants and the yellow object for the other half. Participants were allocated randomly to one of two conditions (Consistent or Inconsistent) in a between-participants design. In the Consistent condition, they heard the labels for each individual object while seeing the same action being performed on this object (e.g. the blue object was always presented as moving up, counterbalanced across participants) across both trials. A second group of participants was allocated to the Inconsistent condition where they saw the same objects and heard the same labels. However, here, participants saw both actions being performed on each object across trials (e.g. the blue object was presented as moving up in one trial and moving to the side in a second trial).

The test phase consisted of eight trials. Each trial presented participants with both images side-by-side on screen as they heard the label for one of these images exactly half way through the trial, i.e.

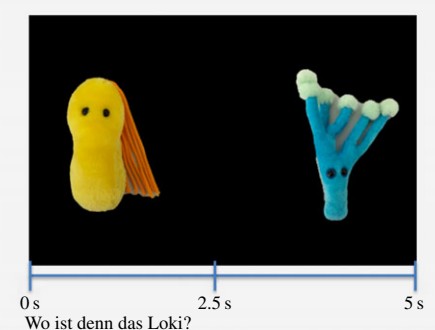

**Figure 3.** Test phase. Both objects are on screen and the target is labelled at 2.5 s. The test phase consisted of eight trials with four trials per label. An attention getter was presented before each trial. All videos are available on OSF under osf.io/tndj7.

2.5 s into the trial, dividing it into a pre- and post-naming phase (figure 3). Gaze points to both images in the pre-naming phase provide an estimate of participants' baseline preference for either of the objects, while gaze points in the post-naming phase indicate participants' response to the relation between the heard label and the images on screen [5].

Overall, the whole procedure lasted 4–5 min on average. Counterbalancing of novel words, actions, and objects resulted in six training lists. Additionally, four test lists were created (2 words × 2 objects, i.e. each object was paired with each label across lists). Presentation order of all trials within blocks was fully randomized.

## 2.5. Preprocessing

The eye tracker provides an estimate of where participants were looking at in each time stamp during the trial, with one data point approximately every 16 ms. All data (gaze data and trial information) were exported from E-Prime and then further processed in R (R v. 3.2.4 (2016-03-10) [38]). For each time stamp, data were only included when one or both eyes of the participant were tracked reliably (validity less than 2 on E-Prime scale). When both eyes were tracked, the mean gaze point for both eyes was computed for further analysis. Gaze data were then aggregated into 40 ms bins. Areas of interest were defined according to the size of the displayed objects and their location on screen.

For the test phase, we coded whether the participant looked at the correct object (i.e. the target), the distractor or at neither of these images on screen. We analysed data that began 240 ms after the onset of the label presentation to ensure that we only analysed eye movements that could be reliably interpreted as a response [39,40]. Furthermore, we subtracted the mean proportion of target looks of the baseline phase (PTL: total looking time at target/(total looking time at target and distractor)) at each 40 ms time point in the post-naming phase on that particular trial.

Single test trials were excluded if a participant looked at the stimuli more or less than $M \mp 2$ s.d. of the trial duration. Thus, a trial was excluded if an 18-month-old child looked 24.1% or less of the time during the trial to one of the two pictures. This led to an exclusion of 48 trials (10.5%). For 30 months, this criterion was at less than or equal to 25.7%, 62 trials (11.4%); for 3- to 4-year-olds it was at less than or equal to 40.6%, 40 trials (9%); and for adults at less than or equal to 37.6%, 28 trials (5.9%). This led to exclusions of some participants (included above) who did not look enough during any of their trials (18: $n = 3$; 30: $n = 3$; 36–48: $n = 1$; adults: $n = 1$; see, for example, [41] for similar values). Furthermore, participants were excluded from the analyses if they provided data from only one trial in the test phase (18: $N = 1$; 30: $N = 1$; 36–48: $N = 1$). This left us with 238 participants ($n$ Consistent/$n$ Inconsistent: 18: 28/26; 30: 33/32; 36–48: 30/29; adults: 30/30).

## 3. Results

Here, we present two different analyses for each age group. In the first analysis, we report traditional ANOVAs and $t$-tests which evaluate our dependent measure, the proportion of fixations to the target collapsed across the post-naming window corrected for fixations to the target in the pre-naming window. In the second analysis, we report generalized linear mixed models. This allows us to include *time* within the trial as an additional factor and examine how children's response develops across the course of the trial, since fixations change rapidly across the length of the time window tested in the current study. Using both of these approaches helps us understand the data from a more differentiated and situated perspective.

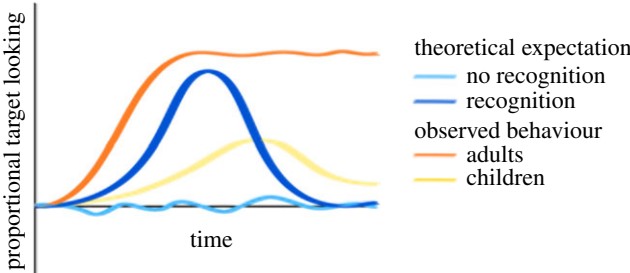

**Figure 4.** Streamer plot of theoretical expectations and typical behavioural observations of looking behaviour across time.

**Table 1.** Descriptives of baseline-corrected proportional target looking in the test phase for the Consistent and Inconsistent condition per age group (18 months, 30 months, 3–4 years and adults). Scores of 0 reflect that averaged target looking is at chance level, i.e. no change from baseline, any values above 0 reflect target looking and values below 0 reflect distractor looking.

| | condition | | | | |
| | Consistent | | | Inconsistent | |
| age group | mean | s.d. | | mean | s.d. |
| --- | --- | --- | --- | --- | --- |
| 18 months | − 0.03 | 0.12 | | 0.00 | 0.19 |
| 30 months | 0.07 | 0.16 | | 0.02 | 0.16 |
| 3–4 years | 0.07 | 0.20 | | 0.03 | 0.17 |
| adults | 0.20 | 0.28 | | 0.23 | 0.29 |

## 3.1. ANOVAs and *t*-tests

For a comparison of conditions across age groups, we ran a 2 (condition) × 4 (age) between-participants ANOVA. For each age group, we ran three *t*-tests: first a two-sample *t*-test comparing the proportion of target looking in the Consistent condition and the Inconsistent condition. Then, we ran separate one-sample *t*-tests comparing the baseline-corrected proportion of fixations to the target in the post-naming window against chance (chance level = 0). Descriptives for each condition per age group can be seen in table 1.

## 3.2. Description of the growth curve model

We also used generalized linear mixed models (GLMM) with *time* as a factor to model children's looking behaviour during the test phase of the experiment (also coined growth curve analysis (GCA); for a more detailed description and instruction see [42,43]). Including time, and its linear, quadratic and cubic polynomial allows us to capture any nonlinear change in looking behaviour across the whole trial duration [42,43]. In figure 4, theoretical expectations and typical looking behaviour are plotted. If the participants do not recognize the target upon hearing the word (i.e. did not associate the word with the target), we expect target looking not to deviate from chance level across the whole time window, which would suggest that participants look more or less randomly at the two objects (light blue). If the participants do recognize the target upon hearing the word, we would expect a quadratic-shaped curve (or quartic if small tails are considered), reflecting how the participant is first at chance level, then looking more to the target upon hearing its label, and then going back to chance level (dark blue). In practice, adults usually stare at the correct object until the end of the trial (orange; see also [42]), which is often better fitted with a linear (steep rise) or cubic function (steep rise and high plateau). Children often show weaker effects in word learning studies with slower rises in the beginning and decreases towards chance level at the end of the trial (yellow), which can often be well-fitted with a cubic polynomial. However, children's target looking appears to be more variable in most cases, so flexibility in fitting the curve as well as in interpreting its course over time are necessary (e.g. [44,45]). Based on these assumptions, we hypothesized that learning in any of the age groups would be reflected on the quadratic or cubic polynomial, and potentially on the linear term for the

adults. If then, participants learn in one condition but not the other, we expect the factor condition to interact with the respective polynomials which would reflect different curves for the two conditions.

We fitted a generalized mixed model using lme4's lmer function in R [46] with Gaussian error structure and identity link function. Condition and time, and their interaction were included as fixed effects of interest. Furthermore, label (Tanu, Löki), object colour (blue object, yellow object), and age at test were included as control factors. Age at test was included in addition to the age group, as the oldest group of children had a larger range in age from 3 to 4 years. We also included Participant id as a random factor to allow for random slopes across participants [43]. Age at test was z-transformed (to a mean of zero and a standard deviation of one) to simplify later interpretation. A reduced model was fit that did not include condition. A comparison between the reduced model and the full model including the factor condition then allows one to evaluate the influence of the factor condition.

As noted above, we also included *time* and its second and third polynomial in the model. This allowed us to model the data as a linear, quadratic and cubic function of time, and thus, to capture according curvatures in the data (the number of polynomials has been restricted to three based on theoretical considerations regarding looking behaviour). An overview of the results can be seen in tables 2 and 3. We used the function drop1 to evaluate the influence of each factor in the model. This function compares the model including one factor with a model without this factor, and thereby evaluates its contribution to the model.

Visual inspection of a qq-plot and a histogram of the residuals showed a normal distribution, but homogeneity appeared to be violated based on a plot of residuals against fitted values. Log-transforming the response did not contribute to an improvement of the model. We, therefore, chose the first model, but results need to be reviewed with care.

## 3.3. All age groups

### 3.3.1. ANOVA

For a comparison of conditions across age groups, we ran a 2 (condition) × 4 (age) ANOVA. We found a significant effect of age ($F_{3,230} = 13.65$, $p < 0.001$, $\eta_p^2 = 0.15$), but no main effect of condition or an interaction of the two factors ($ps > 0.7$). These results suggest that there were differences among age groups but not across conditions or between conditions within age groups.

### 3.3.2. GLMM

Comparing a model including condition with the reduced model without age or condition revealed that condition did not significantly improve the model ($\chi^2 = 3.49$, d.f. = 4, $p = 0.48$). However, including condition and age in the interaction did improve the model significantly compared to a condition-only model ($\chi^2 = 64.06$, d.f. = 24, $p < 0.001$) or an age-only model ($\chi^2 = 27.6$, d.f. = 16, $p = 0.035$). Using drop1, the model revealed a significant interaction of condition × age × poly1 ($\chi^2 = 8.68$, d.f. = 3, $p = 0.034$) and condition × age × poly2 ($\chi^2 = 9.97$, d.f. = 3, $p = 0.019$), suggesting differences between age groups and conditions on the linear and the quadratic term. Thus, these results reflect differences in the time course of target recognition between the two conditions across age groups.

To analyse these effects further, we split the data according to the different age groups. To ensure that the participants truly learned the words in both conditions in the age groups, we further split the data according to condition. The results can be seen in tables 2 and 3.[1]

## 3.4. 18-month-olds

### 3.4.1. *t*-Tests

For the 18-month-olds, a Welch two-sample *t*-test comparing Consistent and Inconsistent conditions found no significant difference between conditions, $p = 0.57$. Separate one-sample *t*-tests comparing baseline-corrected target fixations in each of the conditions to chance level (chance = 0) were not significant (all $ps > 0.1$).

---

[1]Running the same models with a subset of the participants who provided vocabulary information showed that vocabulary knowledge did not influence the results significantly. For comparability across age groups and to preserve a bigger sample size, we focus here on the models without vocabulary information.

**Table 2.** GLMM testing differences between conditions on proportional target looking over time including time, its linear, quadratic and cubic term. res = lmer(PTL_corr.mean condition × age × (poly1 + poly2 + poly3) + object + label + z.TestAge + (1 + (poly1 + poly2 + poly3) | id), data = d_aggr, REML = F, control = contr.

| group | factor | estimates | s.e. | lower CI | upper CI | LRT | p |
|---|---|---|---|---|---|---|---|
| overall | intercept | −0.03 | 0.05 | −0.13 | 0.06 | a | a |
| | object | 0.03 | 0.00 | 0.02 | 0.03 | 65.29 | <0.001 |
| | label | 0.01 | 0.00 | 0.00 | 0.02 | 7.98 | <0.001 |
| | z.TestAge | −0.03 | 0.11 | −0.24 | 0.18 | 0.10 | 0.76 |
| | condition:age:poly1 | a | a | a | a | 8.68 | 0.03 |
| | condition:age:poly2 | a | a | a | a | 9.96 | 0.02 |
| | condition:age:poly3 | a | a | a | a | 3.09 | 0.38 |
| 18 | intercept | −0.09 | 0.13 | −0.35 | 0.18 | a | a |
| | object | 0.04 | 0.01 | 0.02 | 0.05 | 30.72 | <0.001 |
| | label | 0.00 | 0.01 | −0.01 | 0.01 | 0.01 | 0.94 |
| | z.TestAge | −0.22 | 0.39 | −1.02 | 0.53 | 0.32 | 0.57 |
| | condition:poly1 | 0.07 | 0.21 | −0.38 | 0.50 | 0.09 | 0.76 |
| | condition:poly2 | 0.13 | 0.13 | −0.10 | 0.37 | 1.02 | 0.31 |
| | condition:poly3 | −0.00 | 0.10 | −0.21 | 0.19 | 0.00 | 0.97 |
| 30 | intercept | −0.02 | 0.10 | −0.22 | 0.19 | a | a |
| | object | −0.01 | 0.01 | −0.02 | 0.00 | 1.76 | 0.18 |
| | label | −0.01 | 0.01 | −0.02 | 0.00 | 2.59 | 0.11 |
| | z.TestAge | 0.12 | 0.23 | −0.34 | 0.60 | 0.28 | 0.60 |
| | condition:poly1 | −0.07 | 0.19 | −0.44 | 0.29 | 0.15 | 0.70 |
| | condition:poly2 | −0.27 | 0.12 | −0.48 | −0.05 | 4.99 | 0.02 |
| | condition:poly3 | −0.09 | 0.08 | −0.24 | 0.07 | 1.16 | 0.28 |
| 3–4 | intercept | 0.08 | 0.13 | −0.18 | 0.36 | a | a |
| | object | 0.04 | 0.01 | 0.03 | 0.06 | 36.99 | <0.001 |
| | label | 0.05 | 0.01 | 0.04 | 0.06 | 48.70 | <0.001 |
| | z.TestAge | −0.07 | 0.10 | −0.28 | 0.12 | 0.48 | 0.49 |
| | condition:poly1 | 0.50 | 0.21 | 0.06 | 0.87 | 5.36 | 0.02 |
| | condition:poly2 | 0.23 | 0.14 | −0.02 | 0.53 | 2.67 | 0.10 |
| | condition:poly3 | −0.17 | 0.09 | −0.35 | −0.01 | 3.77 | 0.05 |
| adults | intercept | 0.64 | 2.15 | −3.77 | 4.90 | a | a |
| | object | 0.03 | 0.01 | 0.02 | 0.04 | 42.12 | <0.001 |
| | label | −0.01 | 0.01 | −0.02 | 0.00 | 2.07 | 0.15 |
| | z.TestAge | 0.30 | 1.52 | −2.81 | 3.31 | 0.04 | 0.85 |
| | condition:poly1 | −0.28 | 0.17 | −0.61 | 0.05 | 2.80 | 0.09 |
| | condition:poly2 | 0.09 | 0.10 | −0.11 | 0.29 | 0.73 | 0.39 |
| | condition:poly3 | 0.01 | 0.07 | −0.13 | 0.16 | 0.02 | 0.88 |

[a]Note that coefficients of interactions can only be interpreted in relation to the respective baseline levels of the interacting variables. Furthermore, the significance level of intercepts can only be interpreted in a meaningful way when effects on the intercept are tested. Thus, these values are not displayed here because of limited informativity.

### 3.4.2. GLMM

The model comparison between the full and the reduced model was not significant ($\chi^2 = 1.16$, d.f. = 4, $p = 0.89$). When using drop1, no interactions between the conditions and the time terms could be observed. This suggests that condition did not interact with 18-month-olds' target recognition.

**Table 3.** GLMM testing successful learning within conditions over time including time, its linear, quadratic and cubic term. res = lmer(PTL_corr.mean (poly1 + poly2 + poly3) + object + label + z.TestAge + (1 + (poly1 + poly2 + poly3) | id), data = dadult_Inconsistent, REML = F, control = contr.

| group | factor | estimates | s.e. | lower CI | upper CI | LRT | *p* |
|---|---|---|---|---|---|---|---|
| 18 | intercept | 0.16 | 0.27 | −0.41 | 0.70 | a | a |
| Consistent | poly1 | 0.13 | 0.15 | −0.16 | 0.41 | 0.71 | 0.40 |
| | poly2 | 0.15 | 0.09 | −0.01 | 0.34 | 2.75 | 0.10 |
| | poly3 | 0.06 | 0.07 | −0.09 | 0.20 | 0.61 | 0.44 |
| | object | 0.04 | 0.01 | 0.02 | 0.06 | 12.94 | <0.001 |
| | label | 0.04 | 0.01 | 0.02 | 0.06 | 14.40 | <0.001 |
| | z.TestAge | 0.68 | 0.83 | −1.03 | 2.33 | 0.53 | 0.47 |
| 18 | intercept | −0.09 | 0.13 | −0.36 | 0.22 | a | a |
| Inconsistent | poly1 | 0.06 | 0.15 | −0.24 | 0.37 | 0.16 | 0.68 |
| | poly2 | 0.03 | 0.08 | −0.14 | 0.20 | 0.10 | 0.75 |
| | poly3 | 0.06 | 0.07 | −0.07 | 0.19 | 0.68 | 0.41 |
| | object | 0.04 | 0.01 | 0.02 | 0.06 | 17.19 | <0.001 |
| | label | −0.04 | 0.01 | −0.06 | −0.02 | 17.68 | <0.001 |
| | z.TestAge | −0.27 | 0.41 | −1.12 | 0.66 | 0.42 | 0.52 |
| 30 | intercept | 0.17 | 0.16 | −0.16 | 0.50 | a | a |
| Consistent | poly1 | −0.00 | 0.13 | −0.23 | 0.26 | 0.00 | 0.99 |
| | poly2 | −0.23 | 0.08 | −0.38 | −0.07 | 6.80 | 0.01 |
| | poly3 | −0.04 | 0.06 | −0.16 | 0.09 | 0.33 | 0.56 |
| | object | −0.02 | 0.01 | −0.03 | −0.00 | 4.12 | 0.04 |
| | label | −0.04 | 0.01 | −0.05 | −0.02 | 21.04 | <0.001 |
| | z.TestAge | −0.17 | 0.37 | -0.94 | 0.62 | 0.21 | 0.65 |
| 30 | intercept | −0.11 | 0.13 | −0.35 | 0.14 | a | a |
| Inconsistent | poly1 | 0.07 | 0.14 | −0.21 | 0.32 | 0.23 | 0.63 |
| | poly2 | 0.04 | 0.08 | −0.13 | 0.20 | 0.21 | 0.65 |
| | poly3 | 0.05 | 0.05 | −0.05 | 0.16 | 0.90 | 0.34 |
| | object | 0.00 | 0.01 | −0.02 | 0.02 | 0.01 | 0.92 |
| | label | 0.02 | 0.01 | −0.00 | 0.04 | 3.34 | 0.07 |
| | z.TestAge | 0.29 | 0.29 | −0.32 | 0.86 | 0.94 | 0.33 |
| 3–4 | intercept | 0.31 | 0.20 | −0.13 | 0.71 | a | a |
| Consistent | poly1 | 0.36 | 0.14 | 0.08 | 0.61 | 5.92 | 0.01 |
| | poly2 | 0.09 | 0.09 | −0.08 | 0.27 | 0.98 | 0.32 |
| | poly3 | −0.13 | 0.07 | −0.26 | 0.00 | 3.48 | 0.06 |
| | object | 0.02 | 0.01 | 0.00 | 0.04 | 5.07 | 0.02 |
| | label | 0.06 | 0.01 | 0.04 | 0.08 | 37.67 | <0.001 |
| | z.TestAge | −0.22 | 0.15 | −0.52 | 0.11 | 1.67 | 0.20 |
| 3–4 | intercept | −0.07 | 0.17 | −0.44 | 0.27 | a | a |
| Inconsistent | poly1 | −0.14 | 0.16 | −0.45 | 0.17 | 0.83 | 0.36 |
| | poly2 | −0.13 | 0.10 | −0.33 | 0.06 | 1.69 | 0.19 |
| | poly3 | 0.05 | 0.06 | −0.07 | 0.15 | 0.64 | 0.42 |
| | object | 0.07 | 0.01 | 0.05 | 0.09 | 43.65 | <0.001 |

(*Continued.*)

| group | factor | estimates | s.e. | lower CI | upper CI | LRT | p |
|---|---|---|---|---|---|---|---|
| | label | 0.03 | 0.01 | 0.01 | 0.05 | 10.48 | <0.001 |
| | z.TestAge | 0.04 | 0.13 | −0.22 | 0.32 | 0.08 | 0.77 |
| adults | intercept | 2.49 | 2.62 | −2.91 | 8.07 | a | a |
| Consistent | poly1 | 0.16 | 0.13 | −0.09 | 0.42 | 1.50 | 0.22 |
| | poly2 | −0.17 | 0.07 | −0.32 | −0.03 | 5.59 | 0.02 |
| | poly3 | 0.06 | 0.06 | −0.05 | 0.17 | 1.28 | 0.26 |
| | object | 0.04 | 0.01 | 0.03 | 0.06 | 38.64 | 0.001 |
| | label | −0.00 | 0.01 | −0.02 | 0.01 | 0.14 | 0.70 |
| | z.TestAge | 1.64 | 1.86 | −2.20 | 5.56 | 0.71 | 0.40 |
| adults | intercept | −6.49 | 3.55 | −14.20 | 1.22 | a | a |
| Inconsistent | poly1 | 0.44 | 0.11 | 0.22 | 0.64 | 13.80 | <0.001 |
| | poly2 | −0.26 | 0.08 | −0.40 | −0.11 | 9.79 | <0.001 |
| | poly3 | 0.05 | 0.05 | −0.04 | 0.15 | 1.17 | 0.28 |
| | object | 0.02 | 0.01 | 0.01 | 0.04 | 9.29 | <0.001 |
| | label | −0.01 | 0.01 | −0.02 | 0.00 | 1.99 | 0.16 |
| | z.TestAge | −4.73 | 2.51 | −10.13 | 0.71 | 2.76 | 0.10 |

[a]Note that coefficients of interactions can only be interpreted in relation to the respective baseline levels of the interacting variables. Furthermore, the significance level of intercepts can only be interpreted in a meaningful way when effects on the intercept are tested. Thus, these values are not displayed here because of limited informativity.

For neither the model examining the data from the Consistent group alone (Consistent split model), nor the model examining the data from the Inconsistent group alone (Inconsistent split model), did we find significant interactions with the polynomials. Thus, there was no evidence that 18-month-olds had learned and later recognized the word–object mappings in either condition.

## 3.5. 30-month-olds

### 3.5.1. *t*-Tests

For the 30-month-olds, a Welch two-sample *t*-test comparing Consistent and Inconsistent conditions found no significant difference between conditions, $p = 0.22$. Separate one-sample *t*-tests comparing baseline-corrected target fixations in each of the conditions to chance level (chance = 0) showed a significant effect in the Consistent condition ($t_{32} = 2.69$, $p = 0.011$, $d = 0.66$), but not in the Inconsistent condition ($p = 0.4$).

### 3.5.2. GLMM

The model comparison between the full and the reduced model was above the threshold of significance we adopted in the current study ($\chi^2 = 9.26$, d.f. = 4, $p = 0.055$). When using drop1, the model revealed a significant interaction of condition × poly2 ($\chi^2 = 4.99$, d.f. = 1, $p = 0.025$). This result suggests that there are differences in 30-month-olds' target recognition across conditions on the quadratic term.

For the Consistent split model, the model revealed a significant effect on poly2 ($\chi^2 = 6.8$, d.f. = 1, $p = 0.009$). For the Inconsistent split model, we did not find any significant influences of the time terms. Thus, 30-month-olds had learned and later recognized the word–object mappings in the Consistent, but not in the Inconsistent condition.

## 3.6. 3- to 4-year-olds

### 3.6.1. t-Tests

For the 3- to 4-year-olds, a Welch two-sample *t*-test comparing Consistent and Inconsistent conditions found no significant difference between conditions, $p = 0.69$. Separate one-sample *t*-tests comparing

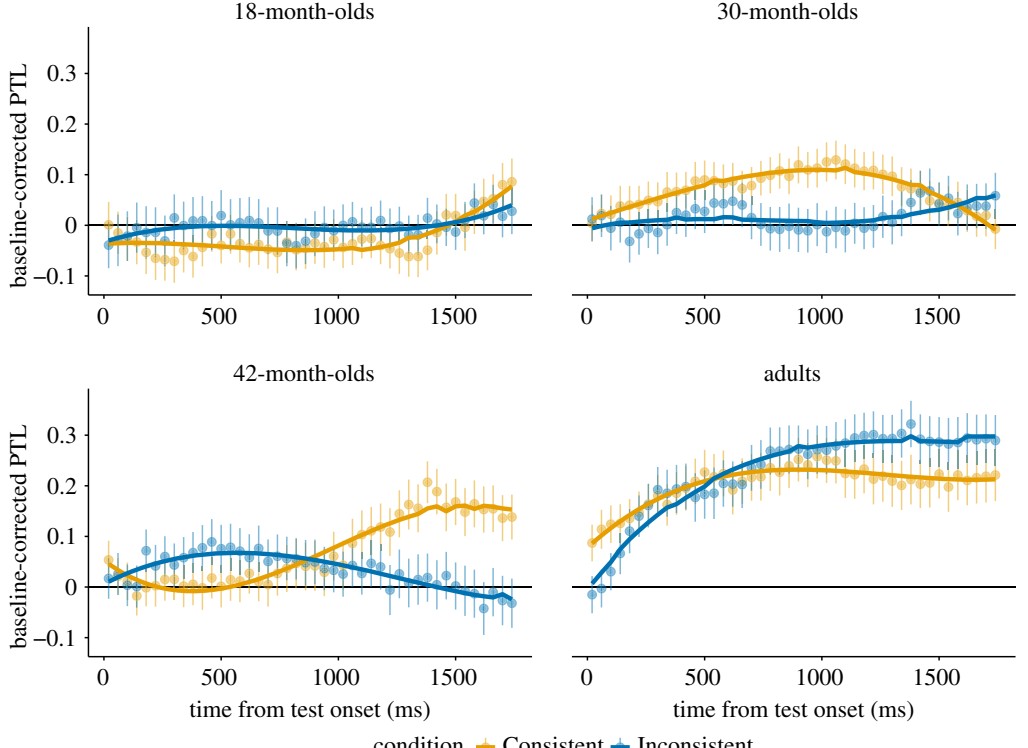

**Figure 5.** Time course graphs for each age group of participants' baseline-corrected proportional target looking (PTL) during the test phase after label onset at 2500 ms and 240 ms to initiate a fixation. The Consistent condition is represented in yellow, the Inconsistent condition is in blue. The line at 0 represents chance level, increases reflect proportionally more target looking whereas decreases reflect distractor looking. The yellow and blue lines reflect the fitted GLMM including time up to the cubic term for each condition.

baseline-corrected target fixations in each of the conditions to chance level (chance = 0) showed no significant effects in either condition (all $ps > 0.1$).

### 3.6.2. GLMM

The model comparison between the full and the reduced model was significant ($\chi^2 = 13.26$, d.f. = 4, $p = 0.01$). Using drop1, there was an interaction between condition × poly1 ($\chi^2 = 5.36$, d.f. = 1, $p = 0.021$) and the interaction of condition × poly3 was above the threshold of significance adopted in the current study ($\chi^2 = 3.77$, d.f. = 1, $p = 0.052$). These results suggest there was a difference in the pattern of 3- to 4-year-olds' target fixations across the two conditions. Visual inspection revealed that in the Consistent condition, children showed a steep rise in fixating the target shortly after the onset of the target label (from around 500 ms) with a peak around 4250 ms. On the contrary, looks to the target in the Inconsistent condition hovered at (or just above) chance throughout the trial (figure 5).

For the Consistent split model, the linear time term was significant ($\chi^2 = 5.92$, d.f. = 1, $p = 0.015$) and the cubic term was above the threshold of significance adopted in the current study ($\chi^2 = 3.48$, d.f. = 1, $p = 0.062$), suggesting that 3- to 4-year-olds had learned the word–object mappings presented to them in the training phase in the Consistent condition, reflected in the sharp increase in their target looking.

For the Inconsistent split model, we did not find any effect on the polynomials. Thus, there was no evidence that 3- to 4-year-olds learned the word–object mapping in the Inconsistent condition.

These results highlight the differences in word learning success between the Consistent and Inconsistent condition at 3- to 4-years of age: target recognition followed different trajectories in the two conditions which is reflected in the linear time term. The split models reveal that only in the Consistent condition, did children learn the word–object associations. This effect was not evident in the t-tests, probably due to the late increase in target looking in the Consistent condition, which resulted in a lower overall mean.

## 3.7. Adults

### 3.7.1. *t*-Tests

For the adults, a Welch two-sample *t*-test comparing Consistent and Inconsistent conditions found no significant difference between conditions, $p = 0.81$. Separate one-sample *t*-tests comparing baseline-corrected target fixations in each of the conditions to chance level (chance = 0) showed a significant effect in the Consistent condition ($t_{29} = 4.02$, $p < 0.001$, $d = 1.04$), and in the Inconsistent condition ($t_{29} = 4.16$, $p < 0.001$, $d = 1.07$).

### 3.7.2. GLMM

The model comparison between the full and the reduced model was not significant ($\chi^2 = 4.44$, d.f. = 4, $p = 0.35$). Using drop1, the model revealed no significant interactions of condition with the time terms. Thus, these results suggest that target looking across time did not differ between conditions for the adults.

For the Consistent split model, the model revealed a significant effect on poly2 ($\chi^2 = 5.59$, d.f. = 1, $p = 0.018$). For the Inconsistent split model, the model revealed a significant effect on poly1 ($\chi^2 = 13.8$, d.f. = 1, $p < 0.001$) and poly2 ($\chi^2 = 9.79$, d.f. = 1, $p = 0.002$).

These results suggest that adults recognized the target successfully in both the Consistent and the Inconsistent conditions. Even for the adults, target recognition followed slightly different trajectories in the two conditions, as can also be seen in figure 5. In the Consistent condition, fixations to the target show a steeper rise and a lower and earlier plateau compared to the Inconsistent condition. However, this difference was not significant in both the *t*-test and the model comparison.

# 4. Discussion

In the current study, we investigated whether the consistency with which particular actions co-occur with particular word–object pairings influences children's word learning. Participants were presented with novel labels for novel objects while these objects either moved in a Consistent (i.e. always the same action across different trials presenting the word–object mapping) or in an Inconsistent (i.e. both actions performed on each object across the different trials presenting the word–object mapping) manner. We did not find any evidence that 18-month-olds learned the word–object mappings in either condition. In contrast, 30-month-olds and 3- to 4-year-olds learned the word–object mappings only in the Consistent condition. This was reflected in the linear, quadratic and cubic time terms of the GLMM, representing the rise and fall of target looking over time in this condition. Only adults learned words for objects in both conditions.

These results suggest that the consistency of co-occurring actions influences 30-month-olds' and 3- to 4-year-old children's word learning. As Gogate *et al.* [4] have shown, actions and words often co-occur in the child's multimodal environment. These actions have also been shown to support word learning [13], especially when provided with temporal synchronicity [1]. However, if these actions appear to be referential, 15-month-olds' word learning is exacerbated [14]. Our results extend these findings in a critical way, showing that it is not just the temporal consistency with which actions accompany word–object associations that impacts word learning success. Between 30 months and 3- to 4-years of age, children learned the word–object mappings only when each word–object mapping had previously been presented with the same action being performed on the object across separate presentations, i.e. in the Consistent condition. This benefit of consistency was not observable at younger ages or with adults. We interpret these findings to suggest that redundant information from different domains supports the forming of rich lexical representations, but only if this information highlights the word–object association and does not distract from it, and only towards the third year of life.

This is in line with the literature suggesting a beneficial effect of consistency on word learning [21], in contrast to a beneficial effect of variability on generalization over different members of a category [29,47,48]. Nevertheless, some recent work suggests an impact of lower-level variability (variability in the colour of the background on which objects were presented) on learning of word–object associations [31]. In contrast to the beneficial effect of variability in the study by Twomey *et al.* no such effect was found in our study. We suggest that the function and salience of the additionally varying information might play a crucial role in whether it boosts or detracts attention to word learning. In Twomey *et al.*'s study, the background colours were a subtle manipulation of the variability in the stimuli. Actions performed on objects, in contrast, may be more salient and compete for attention with the word–object mappings leading to children ignoring the word–object mapping in

favour of increased attention to the actions presented. Increased and salient variability can, therefore, disrupt successful learning in a complex learning environment [29,32]. The results of the current study, thus, highlight the importance of a developmental perspective on the influence of cross-domain information on processing. Taken together with the literature on intersensory redundancy, we trace here a developmental pattern in the influence of actions on word learning. Early in life, with actions that do not detract attention from the linguistic input but rather highlight the association between the linguistic input and objects in the world, co-occurring actions bolster word–object association learning. With increasing age, potentially increasing salience of actions (18 months), and variability in the actions presented (30 months, 3- to 4-years), actions may indeed deter from word learning and lead to children failing to learn the intended word–object mapping. Ongoing work (described in detail below) helps clarify this picture with regard to the salience of actions and words across development.

Dynamic systems and emergentist approaches to development assume that language acquisition and action processing take place in a complex environment where no single aspect of the environment is *a priori* responsible for development in any domain ([49,50], see [51] for an overview). Within this approach, development is the product of active interaction between the child and her immediate environment. As the child develops, she learns to integrate increasing amounts of information while the direct environment adapts to the child's needs and provides the opportunity to learn in a rich and multimodal world.

In their adaptation of such accounts focusing on early word comprehension, Gogate *et al.* [52] suggest that word learning results from the interaction of several processes, including selective attention and intersensory perception, as well as the multimodal input provided to the child (based on [53] and their work on early intermodal perception). Information from different sources and senses regarding the same object might seem unrelated at first, but, when these information sources co-occur, they help to form a full-blown representation. For example, a word and an object might seem arbitrary when presented independently. But presented in temporal synchrony [4] and potentially even with movement of the object [13,54], the relationship between the word and the object is highlighted for the infant, and therefore supports learning. Importantly, perceiving and encoding the relevant aspects of the input requires that the child attends to this input (and indeed, to the relevant aspects of this input). However, attention allocation might be guided by the child's momentary interest in different features of the input. Thus, focusing on a certain type of information can lead to impoverished encoding of other information, which in turn, will shape the spiral staircase of learning in a particular way. Therefore, information from different domains can and does impact processing and learning in the different domains, and can either help or hinder learning across development: only when the child is able to encode all relevant aspects of the complex multimodal input provided to her, will she be able to avoid such momentary pitfalls of attention allocation. Otherwise, learning can be impeded by the complexity of the input that is beyond the child's resources of attention and perception.

With regard to such dynamic accounts of learning, our results suggest that the multimodal complexity of the input interacts with the child's abilities to learn. Multimodal input appears to be helpful in certain learning contexts (e.g. [11]), but a child will only learn what is perceptually salient and relevant in his or her contextual and developmental situation at that specific moment (e.g. [52]). Importantly, we find that the presentation of multimodal input has an impact on word learning, even though this influence is detrimental to word learning at younger ages, especially when co-occurring with salient information from the non-linguistic domain. Our results highlight how children's perception, depending on the age and the developmental context of the child, can be challenged by presenting salient variability in a multimodal context [31], and how this can directly influence learning: words might not be learned as easily if the child is provided with a learning environment that is highly complex, or if that learning environment provides other information that might attract the child's attention. Until age 4, action consistency seems to have a positive cross-domain influence on word learning. In contrast, adults seem robust against the cross-modal influences between word and action learning. Thus, by adulthood we find limited influence of other domain information on word learning.

We note, however, that we cannot conclude from the current study that children devoted more attention to the actions relative to the word–object mappings, since children were presented with words and actions simultaneously [11]. A related study in our laboratory finds, in keeping with these findings, that 12-month-olds do not learn word–object mappings when these are accompanied by actions (see also [14]), while learning to associate actions with objects. Older children (24 and 36 months) do learn word–object mappings but action–object mappings are less successfully learned, suggesting that, at the age where we find an effect of action consistency on word learning, children are able to successfully learn words for objects [45]. Thus, from a certain age, auditorily presented words may be more salient compared to

visually presented actions. While being influenced by the consistency of visually presented actions, children are not distracted by these to the extent that they fail to learn words for objects. However, for the 18-month-olds, we did not find learning in either condition. In accordance with the other study in our laboratory, this could suggest that the visually presented actions here attracted the children's attention to an extent that hindered word learning in the present study. These findings are in contrast to the literature on auditory overshadowing: in these studies, young children usually show an auditory preference, while older children vary in this preference and adults often show a visual preference (see [55,56] for shifts in modality dominance). This developmental trajectory would suggest, in contrast to our suggestions above, that young children focus on auditory words before focusing on visual actions. Thus, our results are difficult to reconcile with those findings and would suggest that auditory dominance develops after the first year. However, the differences in the results could also be due to the types of stimuli presented: object-manipulating actions like the ones used in the present study might appear very salient in contrast to the still images often used in the modality dominance studies [55].

We note also that our findings with 18-month-olds stand in contrast to previous work with younger infants [13] in which even 14-month-olds learned words when these words were presented with objects in motion. In this study, one object (i.e. a dog) was always presented moving to the front of the screen and back while the other (i.e. a toy truck) was presented moving from one side of the screen to the other. Here, children only learned the word–object mappings when the objects were presented in motion but not when they were presented without motion. Notably here, the 14-month-olds were presumably already familiar with images of the dog and the truck and only had to learn the mapping between the novel object and its label, and it is likely that this increased familiarity with the objects may have led to children mapping the words onto these objects with greater ease [57]. The salience of the familiar objects may, therefore, have overridden the effect of salience of the actions on word learning. Furthermore, we note that in some cases, e.g. the study by Gogate and colleagues, infants were presented with three-dimensional objects in a live interaction with their carers [11]. This contrasts with our passive screen-based setting which might have made it more difficult to learn the words (see studies on the video deficit, e.g. [58,59]).

Finally, with regard to the two different statistical approaches, we believe that combining both approaches helped us to quantify the effects we observed. ANOVAs and *t*-tests have been used more often than GLMMs in the word learning literature, and using this approach therefore allowed us to compare our results to previous results in the literature. At the same time, GLMMs which include *time* as a factor provide higher temporal resolution and allow us to include other factors which might be responsible for variance in the data. For example, the GLMMs showed a quartic curve for 18-month-olds' word–object association learning in the Consistent condition, which was absent when target looking was averaged across time. Furthermore, the quartic curve was probably due to a subtle increase towards the end of the trial, and is therefore difficult to reconcile with the theoretical assumptions outlined above. Furthermore, the GLMMs showed a curve for 30-month-olds' word learning in the Consistent condition, and again, this effect was absent when target looking was averaged across time. These differences in results of the 30-month-olds suggest that they did learn the word–object associations in this condition (since target looking followed a predicted curve), but their target looking was only observable when we considered time during the trial to allow for changes in the pattern of looking behaviour across the trial. Even for adults, although they showed learning in both conditions across statistical approaches, the pattern of target looking differed across time during the trial, which might suggest differences in processing. Thus, the combination of ANOVAs, *t*-tests and GLMMs allowed us to present a more differentiated picture of the participants' behaviour. Nonetheless, more research is required with regard to the GLMM and the interpretation of different types of curves.

In conclusion, we find that children's word–object learning between 30 months and 4 years of age was influenced by the consistency with which particular actions co-occur with word–object pairings: in this age range, Consistent word–action–object mappings supported word–object learning in contrast to Inconsistent word–action–object mappings. Only adults learned words independent of the actions in the present setting. In terms of a dynamic systems account, this development reflects how the interaction between the learner and the complexity of the multimodal environment plays a role in shaping the learning experience: it shows how we learn to incorporate parts of a rich multimodal environment that learners of any age experience.

Ethics. Ethics approval was granted by the ethics committee of the Georg-Elias-Müller-Institute for Psychology, University of Goettingen (Project no. 123). Parents signed a written informed consent form for their child. Also adult participants filled out the written consent form.

Data accessibility. Data, code and stimuli videos are available via the Open Science Framework under osf.io/tndj7.
Authors' contributions. N.M. and B.E. had the original research idea and provided funding. S.F.V.E., M.A., B.E. and N.M. designed the study. S.F.V.E. conducted the testings and analysed the data with N.M.'s support. S.F.V.E. and N.M. wrote the manuscript. S.F.V.E., M.A., B.E., and N.M. discussed the results and commented on the final version of the manuscript.
Competing interests. The authors declare that the research was conducted in the absence of any commercial or financial relationships that could be construed as a potential conflict of interest.
Funding. This study was supported by the Deutsche Forschungsgemeinschaft (German Research Foundation), research unit 'Crossing the borders: the interplay of language, cognition, and the brain in early human development' (Project: FOR 2253, grant no. EL 253/7-1). Furthermore, this research was supported by the German Research Foundation (DFG) as part of the RTG 2070 Understanding Social Relationships and the Leibniz-WissenschaftsCampus 'Primate Cognition'. We acknowledge support by the German Research Foundation and the Open Access Publication Funds of the Göttingen University.
Acknowledgements. We thank the families and their children, as well as the adults for participating in the study. Further, we would like to thank Vivien Radtke and Sebastian Isbaner for their helpful comments throughout the process, Roger Mundry and Dan Mirman for statistical support, and Linda Taschenberger, Kristina Goos, Katharina von Zitzewitz and Jonas Reinckens for their help in data collection.

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
