## [Reviewer comments · Royal Society Open Science]

Review History

RSOS-190097.R0 (Original submission)

Review form: Reviewer 1

Is the manuscript scientifically sound in its present form?

Yes

Are the interpretations and conclusions justified by the results?

Yes

Is the language acceptable?

Yes

Is it clear how to access all supporting data?

Yes

Do you have any ethical concerns with this paper?

No

Have you any concerns about statistical analyses in this paper?

Yes

Recommendation?

Accept with minor revision (please list in comments)

Comments to the Author(s)

The manuscript describes a series of studies exploring the effect of action variability on word learning in 18-, 30-, and 36- to 48-month old children and adults. Participants were presented with a looking time task in which they were trained with two novel object-word mappings across four training trials. Participants saw actions performed on the objects during labeling; actions were either the same on each of the two trials (consistent) or different (inconsistent). The youngest children showed no evidence of word learning in either condition. 30-month-old infants and 36- to 48-month old toddlers learned words in the consistent condition only. Finally, adults learned words in both conditions. The authors conclude that young children (but not young infants) can capitalise on redundant multimodal information when learning words.

Note that the supplemental materials on the submission portal appear to relate to bibliographic management; I haven't been able to open them.

Overall the manuscript addresses an interesting question, offers a new contribution to the word learning literature, and will be accessible to a wide audience. I'll first address the points in the reviewer guidelines; I've provided more detail below.

1. Scientific accuracy, including statistical analysis: The use of GCA is relatively novel in infancy research and adds substantial insight into the evolution of children's looking behavior over time. I have some questions about the analyses, detailed below.
2. Whether the research methods are appropriate, and evidence is provided for the conclusions drawn. The research methods are appropriate and the conclusions are merited by the results as they stand, although the Discussion needs some clarification
3. Writing style and accessibility for a wide audience. The paper is largely well-written although there are some places which would benefit from rephrasing. I've listed these in the Minor Points section
4. Use of suitable illustrations, tables and supplementary material to illustrate results. Figures and tables are appropriate and informative.
5. Appropriate length. The manuscript is concise.
6. Ethics. I have no ethical concerns about the research.
7. Data sharing. The authors have made their scripts and data available on Open Science Framework and I have been able to access all files.

Comments and questions

Overall

1) The term semantic (in)consistency suggests a match or mismatch between the semantics of the action and the word; e.g., a hopping action for a toy rabbit is semantically consistent, while a hopping action for a toy snake is semantically inconsistent. Given that stimuli in the current study are novel actions and novel words, there are no a priori semantics, so no match or mismatch between the action/word pairings. "action consistency" (which the authors also use) would be a more neutral term.

Methods

- 2) While it's great that the sample sizes are bigger than is typical in developmental research, it would be helpful to know how these were determined.
- 3) What were the sizes of the AOIs? If AOIs differed substantially, this could explain the main effect of Object which I mention in more detail below.

Results

- 4) p-values of more than .05 should not reported as null, rather than as trends towards significance.
- 5) Although Label and Object were included as control factors, these predictors have a significant effect on the majority of the growth curve models, and the effect of Object in particular is always in the same (positive) direction except for one instance. Although this finding is not central to the motivation of the study, I would like to know what drives it. Are participants only learning names for one of the objects? It would also be clearer to have the same variable names in the tables as in the text (i.e. either Label or Name)
- 6) Table 2: the caption mentions an asterisk but there are no asterisks in the table. Do the authors mean their 1) notation? Relatedly, why are the coefficients for the polynomial interactions not meaningful? I'm not suggesting this is an error - I have limited knowledge of GCM (as would other readers), so a brief explanation might be appropriate (same applies to Table 3).
- 7) Why is sex included as a predictor in the second model (Table 3) but not the first (Table 2)? Looking at the R script I think this is a typo in the MS.
- 8) It's not clear from the participants section that only a subset of parents completed the language measures. In the footnote on page 9 the authors explain that the language measures were dropped in order to preserve sample size. On my understanding, though, LMCMs can deal with missing data, so it would seem sensible to report models including these measures unless the subset is very small. Alternatively, if the language measures had no statistical effect, the authors could state this in the footnote - "did not contribute strongly to the models" is ambiguous as to significance.

Discussion

- 9) In places I found the flow of the discussion difficult to follow: dynamic systems approaches and multimodal input are introduced on page 12, followed by the Gogate/Hollich Emergentist Coalition Model, followed by auditory overshadowing (see my next comment on this section), followed by dynamic systems and multimodal input, followed by dynamic systems again. The authors also bring in active learning; although this is an important field I feel mentioning it here muddies the water, as this study used a non-naturalistic design. Overall, subheadings would help to make the line of argument clearer
- 10) On pages 12-13 the authors talk about salience of language over actions and argue that "from a certain age" words are more salient than actions. Are they referring to their 30-month-olds here, and is the claim that language is less salient for younger children? How does this relate to the auditory overshadowing effects shown in infants (e.g., Robinson & Sloutsky, 2007)? Given the relevance of the auditory overshadowing literature to the discussion in this paragraph it would be helpful to have more detail.

Minor points

Abstract:

“exacerbate” -> “hinder”

Ages: present ranges for all age groups, or months for all age groups; be consistent with use of months versus years

p.2, l.34: “Temporal alignment of actions accompanying caregiver speech is around 76% during year 1 [4] and have already been shown to accompany 2-year-old children’s speech [10].” This sentence isn’t clear; perhaps “...and 2-year-old children accompany their own speech with actions”

p.2, l.42: “Young infants seem to benefit from multimodal input when, for example, they were presented” – tenses are mixed here

p.2, l.56: an example of what these authors considered a referential action to be would be helpful

p.3, l.10: “Language can also facilitate the comparison of actions already in 10-month-olds” – “in infants as young as 10 months” would work better than “already in 10-month-olds” (similar comment applies to uses of “already in X-month-olds” elsewhere)

p.3, l.30: “studies have shown that consistency in form of the same location associations” -> “studies have shown that consistency in referent location”

p.3, l.54: “although 30-month-olds would learn labels” -> “although 30-month-olds learned labels”

p.4, l.27: “passed” -> “past”

p.4, l.28: “does not influence” -> “would not influence”

p. 6, l.46: “For the test phase, it was coded” -> “For the test phase, we coded”

p.7, l.14 “childrenâAZ s”: something’s gone wrong with formatting here

p.7, l.50: “If, however, they look to the correct target during the time” – I wasn’t clear what this sentence meant

Figure 4: “Time from label onset” would be clearer as the figure caption

p.12, l.35: “a word and an object might seem arbitrary”: this isn’t clear; perhaps “a word and an object might seem unrelated”

p.12, l.58: “auditory words, and language in particular, may be more salient”: this suggests that language is part of words, rather than words being part of language... “words may be more salient” would work better.

p.13, l. 9: “Of interest is also the difference” -> “Also of interest is the difference”

p.13, l.21: the mention of the water wheel comes out of the blue

Review form: Reviewer 2

Is the manuscript scientifically sound in its present form?

No

Are the interpretations and conclusions justified by the results?

Yes

Is the language acceptable?

Yes

Is it clear how to access all supporting data?

No

Do you have any ethical concerns with this paper?

No

Have you any concerns about statistical analyses in this paper?

Yes

Recommendation?

Major revision is needed (please make suggestions in comments)

Comments to the Author(s)

See document attached (Appendix A).

Decision letter (RSOS-190097.R0)

28-May-2019

Dear Ms Eiteljoerge,

The editors assigned to your paper ("Semantic consistency of actions influences young children's word learning") have now received comments from reviewers. We would like you to revise your paper in accordance with the referee and Associate Editor suggestions which can be found below (not including confidential reports to the Editor). Please note this decision does not guarantee eventual acceptance.

Please submit a copy of your revised paper before 20-Jun-2019. Please note that the revision deadline will expire at 00.00am on this date. If we do not hear from you within this time then it will be assumed that the paper has been withdrawn. In exceptional circumstances, extensions may be possible if agreed with the Editorial Office in advance. We do not allow multiple rounds of revision so we urge you to make every effort to fully address all of the comments at this stage. If deemed necessary by the Editors, your manuscript will be sent back to one or more of the original reviewers for assessment. If the original reviewers are not available, we may invite new reviewers.

If your study uses humans or animals please include details of the ethical approval received, including the name of the committee that granted approval. For human studies please also detail

whether informed consent was obtained. For field studies on animals please include details of all permissions, licences and/or approvals granted to carry out the fieldwork.

- Data accessibility

If you wish to submit your supporting data or code to Dryad (<http://datadryad.org/>), or modify your current submission to dryad, please use the following link:
<http://datadryad.org/submit?journalID=RSOS&manu=RSOS-190097>

- Competing interests

- Authors' contributions

- Acknowledgements

- Funding statement

Kind regards,
Andrew Dunn
Royal Society Open Science Editorial Office

on behalf of Dr Shirley-Ann Rüschemeyer (Associate Editor) and Essi Viding (Subject Editor)
openscience@royalsociety.org

Associate Editor's comments (Dr Shirley-Ann Rüschemeyer):

Associate Editor: 1

Comments to the Author:

Dear Authors,

Thank you for submitting your work to the Royal Society Open Science journal for consideration. I apologize for the amount of time it has taken to provide you with an evaluation – it took longer than usual to find reviewers for your manuscript. I'm pleased to tell you that on the basis of my own assessment of the manuscript, as well as the comments of two reviewers, we would like to accept the manuscript for publication pending revisions. There are two major things that should be addressed in your revision:

1. Clarification of Methods/Results:

- I am confused about what the relative contribution of the two analyses you present are: in particular, I do not understand what the ANOVA/t-tests add to your interpretation of your data. This is particularly so as the overall ANOVA shows no significant interaction between Age x Condition, rendering follow-up age-specific t-tests unwarranted. This stands in contrast to what you see in the more powerful GLMM analysis. Please could you clarify what we learn from the ANOVAs? It's not clear to me that they are needed at all.
- Reviewer 1 notes a number of other issues in the reporting of methods/results, all of which I think can be dealt with relatively easily.

2. Both reviewers also note problems with drawing comparisons between the action consistencies manipulated in your paradigm and "action semantics". Reviewer 1 has suggested that much of the problem could be eliminated by simply talking about "action consistency" rather than semantic information; this is also my impression. This opens up avenues for discussion about how the presence of action features may contribute to semantic representations (i.e., consistent actions result in an action feature, inconsistent action pairings do not).

The reviewers also bring up a number of more minor points, which you may consider if you feel it improves your manuscript. In general, my advice would be to focus on the substantive points raised, rather than focusing on typos.

The revised manuscript will be sent back to at least one reviewer.

Best wishes,
Shirley-Ann Rueschemeyer

Comments to Author:

Reviewers' Comments to Author:

Reviewer: 1

Comments to the Author(s)

The manuscript describes a series of studies exploring the effect of action variability on word learning in 18-, 30-, and 36- to 48-month old children and adults. Participants were presented

with a looking time task in which they were trained with two novel object-word mappings across four training trials. Participants saw actions performed on the objects during labeling; actions were either the same on each of the two trials (consistent) or different (inconsistent). The youngest children showed no evidence of word learning in either condition. 30-month-old infants and 36- to 48-month old toddlers learned words in the consistent condition only. Finally, adults learned words in both conditions. The authors conclude that young children (but not young infants) can capitalise on redundant multimodal information when learning words.

Note that the supplemental materials on the submission portal appear to relate to bibliographic management; I haven't been able to open them.

Overall the manuscript addresses an interesting question, offers a new contribution to the word learning literature, and will be accessible to a wide audience. I'll first address the points in the reviewer guidelines; I've provided more detail below.

1. Scientific accuracy, including statistical analysis: The use of GCA is relatively novel in infancy research and adds substantial insight into the evolution of children's looking behavior over time. I have some questions about the analyses, detailed below.
2. Whether the research methods are appropriate, and evidence is provided for the conclusions drawn. The research methods are appropriate and the conclusions are merited by the results as they stand, although the Discussion needs some clarification
3. Writing style and accessibility for a wide audience. The paper is largely well-written although there are some places which would benefit from rephrasing. I've listed these in the Minor Points section
4. Use of suitable illustrations, tables and supplementary material to illustrate results. Figures and tables are appropriate and informative.
5. Appropriate length. The manuscript is concise.
6. Ethics. I have no ethical concerns about the research.
7. Data sharing. The authors have made their scripts and data available on Open Science Framework and I have been able to access all files.

Comments and questions

Overall

1) The term semantic (in)consistency suggests a match or mismatch between the semantics of the action and the word; e.g., a hopping action for a toy rabbit is semantically consistent, while a hopping action for a toy snake is semantically inconsistent. Given that stimuli in the current study are novel actions and novel words, there are no a priori semantics, so no match or mismatch between the action/word pairings. "action consistency" (which the authors also use) would be a more neutral term.

Methods

2) While it's great that the sample sizes are bigger than is typical in developmental research, it would be helpful to know how these were determined.

3) What were the sizes of the AOIs? If AOIs differed substantially, this could explain the main effect of Object which I mention in more detail below.

Results

4) p-values of more than .05 should not reported as null, rather than as trends towards significance.

5) Although Label and Object were included as control factors, these predictors have a significant effect on the majority of the growth curve models, and the effect of Object in particular is always in the same (positive) direction except for one instance. Although this finding is not central to the motivation of the study, I would like to know what drives it. Are participants only learning names for one of the objects? It would also be clearer to have the same variable names in the tables as in the text (i.e. either Label or Name)

6) Table 2: the caption mentions an asterisk but there are no asterisks in the table. Do the authors mean their 1) notation? Relatedly, why are the coefficients for the polynomial interactions not meaningful? I'm not suggesting this is an error - I have limited knowledge of GCA (as would other readers), so a brief explanation might be appropriate (same applies to Table 3).

7) Why is sex included as a predictor in the second model (Table 3) but not the first (Table 2)? Looking at the R script I think this is a typo in the MS.

8) It's not clear from the participants section that only a subset of parents completed the language measures. In the footnote on page 9 the authors explain that the language measures were dropped in order to preserve sample size. On my understanding, though, LMEMs can deal with missing data, so it would seem sensible to report models including these measures unless the subset is very small. Alternatively, if the language measures had no statistical effect, the authors could state this in the footnote - "did not contribute strongly to the models" is ambiguous as to significance.

Discussion

9) In places I found the flow of the discussion difficult to follow: dynamic systems approaches and multimodal input are introduced on page 12, followed by the Gogate/Hollich Emergentist Coalition Model, followed by auditory overshadowing (see my next comment on this section), followed by dynamic systems and multimodal input, followed by dynamic systems again. The authors also bring in active learning; although this is an important field I feel mentioning it here muddies the water, as this study used a non-naturalistic design. Overall, subheadings would help to make the line of argument clearer

10) On pages 12-13 the authors talk about salience of language over actions and argue that "from a certain age" words are more salient than actions. Are they referring to their 30-month-olds here, and is the claim that language is less salient for younger children? How does this relate to the auditory overshadowing effects shown in infants (e.g., Robinson & Sloutsky, 2007)? Given the relevance of the auditory overshadowing literature to the discussion in this paragraph it would be helpful to have more detail.

Minor points

Abstract:

"exacerbate" -> "hinder"

Ages: present ranges for all age groups, or months for all age groups; be consistent with use of months versus years

p.2, l.34: "Temporal alignment of actions accompanying caregiver speech is around 76% during year 1 [4] and have already been shown to accompany 2-year-old children's speech [10]." This

sentence isn't clear; perhaps "...and 2-year-old children accompany their own speech with actions"

p.2, l.42: "Young infants seem to benefit from multimodal input when, for example, they were presented" – tenses are mixed here

p.2, l.56: an example of what these authors considered a referential action to be would be helpful

p.3, l.10: "Language can also facilitate the comparison of actions already in 10-month-olds" – "in infants as young as 10 months" would work better than "already in 10-month-olds" (similar comment applies to uses of "already in X-month-olds" elsewhere)

p.3, l.30: "studies have shown that consistency in form of the same location associations" -> "studies have shown that consistency in referent location"

p.3, l.54: "although 30-month-olds would learn labels" -> "although 30-month-olds learned labels"

p.4, l.27: "passed" -> "past"

p.4, l.28: "does not influence" -> "would not influence"

p. 6, l.46: "For the test phase, it was coded" -> "For the test phase, we coded"

p.7, l.14 "childrenâAZ s": something's gone wrong with formatting here

p.7, l.50: "If, however, they look to the correct target during the time" – I wasn't clear what this sentence meant

Figure 4: "Time from label onset" would be clearer as the figure caption

p.12, l.35: "a word and an object might seem arbitrary": this isn't clear; perhaps "a word and an object might seem unrelated"

p.12, l.58: "auditory words, and language in particular, may be more salient": this suggests that language is part of words, rather than words being part of language... "words may be more salient" would work better.

p.13, l. 9: "Of interest is also the difference" -> "Also of interest is the difference"

p.13, l.21: the mention of the water wheel comes out of the blue

Reviewer: 2

Comments to the Author(s)

See document attached

Author's Response to Decision Letter for (RSOS-190097.R0)

See Appendix B.

Decision letter (RSOS-190097.R1)

02-Jul-2019

Dear Ms Eiteljoerge,

I am pleased to inform you that your manuscript entitled "Consistency of co-occurring actions influences young children's word learning" is now accepted for publication in Royal Society Open Science.

Kind regards,

on behalf of Dr Shirley-Ann Rüschemeyer (Associate Editor) and Essi Viding (Subject Editor)
openscience@royalsociety.org

Associate Editor Comments to Author (Dr Shirley-Ann Rüschemeyer):

Dear Authors,

Thank you for thoroughly addressing the reviewer's comments and concerns. I am happy to accept your revision for publication in Royal Society Open Science.

Many thanks again for submitting your work to RSOS.

Best wishes,

Follow Royal Society Publishing on Twitter: [@RSocPublishing](https://twitter.com/RSocPublishing)
Follow Royal Society Publishing on Facebook:
<https://www.facebook.com/RoyalSocietyPublishing.FanPage/>
Read Royal Society Publishing's blog: <https://blogs.royalsociety.org/publishing/>

Appendix A

Here are my comments, I certainly believe your work has merit. I particularly like the growth model curve analyses, which make a lot of sense developmentally. However in my analysis that Some major revisions are necessary:

Using zscores and median when comparing children and adults for first ANOVA. I also did not see anything in the results or discussion on the particular point of residual analysis that violated the homogeneity constraint, nor whether you had any power analysis results.

Clarifying hand guided movements and representational actions (gestures), removing the rabbit example.

Situating your work compared to Gogate's, you are not doing Pucini et al.'s (gestures)

Adding new hypotheses in the discussion.

Below are in-text my comments:

Introduction

Page 2

Line 12: this sentence seems like an overgeneralization to me. Depending on context, individual differences, samples, the numbers will change. I suggest to add 'in the context of the studies of XYZ', ...

Is this a new method? Is this a modification of a method? Please reference in first paragraph!

Line 29: we do not know if the pace are similar, generalization here. May be say that they also rapidly learn actions...

Lines 30-33: it is one study with star fish shape only (not very ecologically valid) and p at .048. So may be say that a first study using star fish actions suggest that 7 to 9 month olds preferred looking at... than at... , which the authors interpreted as statistical learning.

Line 35: again, overgeneralization, rephrase.

Line 54-56: in the study cited, they used representational gestures (symbolic) not movements like Gogate's. It should be distinguished throughout the paper whether you are talking about gestures or actions or it gets confusing. May be define the terminology you are going to use: actions, movements, gestures...

Page 3

First paragraph: use 'may' rather 'can ' or results suggest that under certain circumstances, children can...

Do you consider your 'actions' Hopping and wiggling ears for rabbits, just as gestures (conventionalized ones) ?

Justification of choice. There is a confusion here for the reader as the experimenter move the tani and löcki whole body = activated movements but the example (hopping and wiggling)

provided is about gestures. May be use Gogate examples, that are also about activated movements by a hand.

Line 44 and next: Presenting variability separated from consistency is one way to do it, however literature shows that both are necessary and possibly in interaction, may be suggest this. I believe it is is what you did as you have a within design.

Page 4:

Results should not be mixed with hypotheses and so remove line 30-31 and Add a paragraph on why you selected 18 month olds, and 30 month olds and 3 to 4 year olds based on the literature.

I believe it should be whom not who.

Page 5 methods

Cite reference for preferential looking paradigm test applied to language and intermodal (PLP) Hirsh-Pasek, K., & Golinkoff, R. M. (1996). The intermodal preferential looking paradigm: A window onto emerging language comprehension. In D. McDaniel, C. McKee, & H. S. Cairns (Eds.), *Language, speech, and communication. Methods for assessing children's syntax* (pp. 105-124). Cambridge, MA, US: The MIT Press.

It is certainly relevant for 18 months old, reference needed for 3 to 4 year-olds.

Page 6

How long did the whole procedure last? The training, the testing the

Experimental design: a graph of the experimental design would help here with the time line, the yoke training, the testing, the attention getter, etc.

Line 52-53. Is this common procedure? Are the % of trials excluded comparable to other studies? Please cite references.

Page 7

Line 14 typo

Design: line 11 please report that it is a between participants design. Anova, etc ... your DV is preferential target looking collapsed over time.

Please indicate if you had power measure

You are testing consistency and variability independently but children are also exposed to those

concomitantly, why did you not consider this, justify.

Table 1, what is the unit of measure?

Why is the mean 000, I do not understand these numbers, are they zscores?

Page 8

Growth model curve

Line 53. You start with a theoretical model and then go to 'in practice'. Do you have any references to provide for the 'in practice'? for adults and children? I like the graph, very useful.

Add a reference for adding a participant id as random factor.

The controls are well thought of and explained.

As are the linear, quadratic and cubic function of time and also target fixation patterns.

Homogeneity violated in residuals.

Line 49 ? power calculations undertaken to ensure that sample sizes were adequate to test the hypothesis considered; are these zscores? They should be, may be use median rather than mean for outliers in zscores.

The results are well explained and provide an interesting developmental comparison.

It would be important to check the results with your entire data [including the removed data] to see if what you had removed would change the results.

Discussion

In Gogate, the setting was quite different a real world situation, with a mom and a 3D object in a meaningful a playful setting. So comparisons may not be granted here, and is another hypothesis as to why you did not find an effect besides the background hypothesis in younger children.

You may want to cite references having found issue with learning from screen, TV, ... in toddlers maybe P. Kuhl: learning from social experience?

Line 45: Children also learn word independent of action, these are called conventions!
Overgeneralization.

Line 46: you cannot use stage with dynamic systems, it is antinomic. You can use developmental time instead (because of individual differences).

Line 45

Appendix B

Review: Semantic consistency of actions influences young children's word learning

Associate Editor

Thank you for submitting your work to the Royal Society Open Science journal for consideration. I apologize for the amount of time it has taken to provide you with an evaluation—it took longer than usual to find reviewers for your manuscript. I'm pleased to tell you that on the basis of my own assessment of the manuscript, as well as the comments of two reviewers, we would like to accept the manuscript for publication pending revisions. There are two major things that should be addressed in your revision:

1. Clarification of Methods/Results:

- I am confused about what the relative contribution of the two analyses you present are: in particular, I do not understand what the ANOVA/t-tests add to your interpretation of your data. This is particularly so as the overall ANOVA shows no significant interaction between Age x Condition, rendering follow-up age-specific t-tests unwarranted. This stands in contrast to what you see in the more powerful GLMM analysis. Please could you clarify what we learn from the ANOVAs? It's not clear to me that they are needed at all.
- Reviewer 1 notes a number of other issues in the reporting of methods/results, all of which I think can be dealt with relatively easily.
 - Thank you for this comment. We included both types of analyses because we believe that the ANOVAs and t-tests make it easier to situate our results in the broader context of the literature, while the GLMMs provide higher resolution and allow us to account for variance in the data. Further, we believe that comparisons between the two analyses allow for a more differentiated picture of the results. We have now included a paragraph on the statistical approaches in the discussion.

2. Both reviewers also note problems with drawing comparisons between the action consistencies manipulated in your paradigm and "action semantics". Reviewer 1 has suggested that much of the problem could be eliminated by simply talking about "action consistency" rather than semantic information; this is also my impression. This opens up avenues for discussion about how the presence of action features may contribute to semantic representations (i.e., consistent actions result in an action feature, inconsistent action pairings do not).

- We have now used "consistency" instead of "semantic consistency" throughout the manuscript.

The reviewers also bring up a number of more minor points, which you may consider if you feel it improves your manuscript. In general, my advice would be to focus on the substantive points raised, rather than focusing on typos.

Reviewer 1

The manuscript describes a series of studies exploring the effect of action variability on word learning in 18-, 30-, and 36- to 48-month old children and adults. Participants were presented with a looking time task in which they were trained with two novel object-word mappings across four training trials. Participants saw actions performed on the objects during labeling; actions were either the same on each of the two trials (consistent) or different (inconsistent). The youngest children showed no evidence of word learning in either condition. 30-month-old infants and 36- to 48-month old toddlers learned words in the consistent condition only. Finally, adults learned words in both conditions. The authors conclude that young children (but not young infants) can capitalise on redundant multimodal information when learning words.

Note that the supplemental materials on the submission portal appear to relate to bibliographic management; I haven't been able to open them.

- Thank you for this comment. We checked this again to ensure that this was the .bib file that is required to compile a LaTeX document with bibliography entries.

Overall the manuscript addresses an interesting question, offers a new contribution to the word learning literature, and will be accessible to a wide audience. I'll first address the points in the reviewer guidelines; I've provided more detail below.

- Thank you very much for your detailed feedback! We highly appreciate your comments and believe that they have significantly improved our work and the manuscript.

1. Scientific accuracy, including statistical analysis: The use of GCA is relatively novel in infancy research and adds substantial insight into the evolution of children's looking behavior over time. I have some questions about the analyses, detailed below.
2. Whether the research methods are appropriate, and evidence is provided for the conclusions drawn. The research methods are appropriate and the conclusions are merited by the results as they stand, although the Discussion needs some clarification
3. Writing style and accessibility for a wide audience. The paper is largely well-written although there are some places which would benefit from rephrasing. I've listed these in the Minor Points section
4. Use of suitable illustrations, tables and supplementary material to illustrate results. Figures and tables are appropriate and informative.
5. Appropriate length. The manuscript is concise.
6. Ethics. I have no ethical concerns about the research.
7. Data sharing. The authors have made their scripts and data available on Open Science Framework and I have been able to access all files.

Comments and questions

Overall

1) The term semantic (in)consistency suggests a match or mismatch between the semantics of the action and the word; e.g., a hopping action for a toy rabbit is semantically consistent, while a hopping action for a toy snake is semantically inconsistent. Given that stimuli in the current study are novel actions and novel words, there are no a priori semantics, so no match or mismatch between the action/word pairings. “action consistency” (which the authors also use) would be a more neutral term.

- We now use “consistency” instead of “semantic consistency” throughout the manuscript.

Methods

2) While it’s great that the sample sizes are bigger than is typical in developmental research, it would be helpful to know how these were determined.

- There are few comparable studies looking at children’s learning of words with simultaneously presented actions. Studies do exist with regard to the influence of variability on learning but these appear to be underpowered (e.g., Twomey, Ma & Westermann, 2017). We therefore did not have a suitable prior on which to base the power analyses. We therefore decided to test a higher number of children than usual relative to earlier studies, aiming for at least 24 children per condition per age-group.

3) What were the sizes of the AOIs? If AOIs differed substantially, this could explain the main effect of Object which I mention in more detail below.

- The AOIs were identical in size and we have now added this information in the methods.

Results

4) p-values of more than .05 should not be reported as null, rather than as trends towards significance.

- We now report that these values are above the threshold we set for significance in our study.

5) Although Label and Object were included as control factors, these predictors have a significant effect on the majority of the growth curve models, and the effect of Object in particular is always in the same (positive) direction except for one instance. Although this finding is not central to the motivation of the study, I would like to know what drives it. Are participants only learning names for one of the objects? It would also be clearer to have the same variable names in the tables as in the text (i.e. either Label or Name)

- As you note, we believe it to be important that these factors were included as control factors. Based on a visual analysis of the data we do not think it is the case that participants learned names for only one of the objects. Rather, we believe these differences arise from differences in the curvature of the response to both objects at best. For comparison we plotted the data for all age groups in the consistent condition separated by objects here and provide the code on OSF.

- We have now changed the wording in the table.

6) Table 2: the caption mentions an asterisk but there are no asterisks in the table. Do the authors mean their 1) notation? Relatedly, why are the coefficients for the polynomial interactions not meaningful? I'm not suggesting this is an error - I have limited knowledge of GCM (as would other readers), so a brief explanation might be appropriate (same applies to Table 3).

- Thank you for this remark. Coefficients reflect the amount of influence of one specific level of a factor. For example, we might have the factor "Object" with the levels "blue" and "yellow". The GLMM automatically assigns a "baseline level", let's say "blue" in this example (usually ordered by alphabet or some other order indicator). A main effect of Object can then be seen as a change in the coefficient for the yellow object compared to the blue object. When we now consider an interaction, let's say between the factor "Object" and "Label", this coefficient would reflect the change for object (i.e., change for yellow compared to blue) and for label (i.e., Tanu compared to Loki) and their interaction. Accordingly, the coefficients do not represent the interaction alone, and it is therefore not helpful to interpret them with regard to the interaction. We have now included a note on this in the table footnote.

7) Why is sex included as a predictor in the second model (Table 3) but not the first (Table 2)? Looking at the R script I think this is a typo in the MS.

- Thank you for noting this, it was indeed a typo and we corrected the description of the model now.

8) It's not clear from the participants section that only a subset of parents completed the language measures. In the footnote on page 9 the authors explain that the language measures were dropped in order to preserve sample size. On my understanding, though, LMEMs can deal with missing data, so it would seem sensible to report models including

these measures unless the subset is very small. Alternatively, if the language measures had no statistical effect, the authors could state this in the footnote - "did not contribute strongly to the models" is ambiguous as to significance.

- Vocabulary scores did not significantly influence the results. We have changed the phrasing accordingly. Because the function *drop1* cannot deal with missing data but provides a good method to evaluate the main effects and interactions, we decided to keep the models without the vocabulary score. Note however, that they are part of the dataset on OSF.

Discussion

9) In places I found the flow of the discussion difficult to follow: dynamic systems approaches and multimodal input are introduced on page 12, followed by the Gogate/Hollich Emergentist Coalition Model, followed by auditory overshadowing (see my next comment on this section), followed by dynamic systems and multimodal input, followed by dynamic systems again. The authors also bring in active learning; although this is an important field I feel mentioning it here muddies the water, as this study used a non-naturalistic design. Overall, subheadings would help to make the line of argument clearer

- Thank you for this comment. We have now reorganized the discussion and hope that the argumentation is now clearer.

10) On pages 12-13 the authors talk about salience of language over actions and argue that "from a certain age" words are more salient than actions. Are they referring to their 30-month-olds here, and is the claim that language is less salient for younger children? How does this relate to the auditory overshadowing effects shown in infants (e.g., Robinson & Sloutsky, 2007)? Given the relevance of the auditory overshadowing literature to the discussion in this paragraph it would be helpful to have more detail.

- This is a very important point. We have now extended this section and discuss our results with regard to auditory overshadowing in more detail.

Minor points

Abstract:

"exacerbate" -> "hinder"

- We have now changed this sentence to "This type of additional information in the context might influence young children's word learning." to incorporate all possible directions in which the influence could go.

Ages: present ranges for all age groups, or months for all age groups; be consistent with use of months versus years

- We have now changed it to months for all age groups.

p.2, l.34: "Temporal alignment of actions accompanying caregiver speech is around 76% during year 1 [4] and have already been shown to accompany 2-year-old children's speech [10]." This sentence isn't clear; perhaps "...and 2-year-old children accompany their own speech with actions"

- We have now included your version of the sentence in the manuscript.

p.2, l.42: “Young infants seem to benefit from multimodal input when, for example, they were presented” – tenses are mixed here

- We have now changed the beginning of this paragraph: “The high co-occurrence of words and actions in the input of the child might be accompanied by cross-domain influences between words and actions. Indeed, studies have shown that young infants seem to benefit from multimodal input when learning novel words.” We then continue with the review of the literature.

p.2, l.56: an example of what these authors considered a referential action to be would be helpful

- We have now added an example from their paper.

p.3, l.10: “Language can also facilitate the comparison of actions already in 10-month-olds” – “in infants as young as 10 months” would work better than “already in 10-month-olds” (similar comment applies to uses of “already in X-month-olds” elsewhere)

- We have now changed both sentences.

p.3, l.30: “studies have shown that consistency in form of the same location associations” -> “studies have shown that consistency in referent location”

- We have now changed this sentence accordingly.

p.3, l.54: “although 30-month-olds would learn labels” -> “although 30-month-olds learned labels”

- We have now changed this sentence accordingly.

p.4, l.27: “passed” -> “past”

- We have now replaced “passed” with “past”.

p.4, l.28: “does not influence” -> “would not influence”

- We have now replaced “does” with “would”.

p. 6, l.46: “For the test phase, it was coded” -> “For the test phase, we coded”

- We have now changed this sentence accordingly.

p.7, l.14 “childrenâ~AZ´ s”: something’s gone wrong with formatting here

- We fixed the formatting.

p.7, l.50: “If, however, they look to the correct target during the time” – I wasn’t clear what this sentence meant

- We rephrased this sentence and hope it is now clearer: “If the participants do recognise the target upon hearing its label, we would expect a quadratic- or quartic-shaped curve, reflecting how participants’ fixations are initially at chance level (prior to hearing the label), increasing towards to the target upon hearing its label, and then going back to chance level (dark blue).”

Figure 4: “Time from label onset” would be clearer as the figure caption

- We thought that “time from label onset” would be misleading since we exclude the data up to 240 ms after label onset to allow for the initiation of a fixation (e.g., Swingley et al., 1999).

p.12, l.35: “a word and an object might seem arbitrary”: this isn’t clear; perhaps “a word and an object might seem unrelated”

- We have now changed this sentence accordingly.

p.12, l.58: “auditory words, and language in particular, may be more salient”: this suggests that language is part of words, rather than words being part of language... “words may be more salient” would work better.

- We have now changed this sentence accordingly.

p.13, l. 9: “Of interest is also the difference” -> “Also of interest is the difference”

- We have now changed this sentence accordingly.

p.13, l.21: the mention of the water wheel comes out of the blue

- We have rephrased this sentence now by leaving the water wheel out and focusing on the contrast between familiar and novel objects: “Notably here, the 14-month-olds were presumably already familiar with images of the dog and the truck and only had to learn the mapping between the novel object and its label, and it is likely that this increased familiarity with the objects may have led to children mapping the words onto these objects with greater ease”.

Reviewer 2

Here are my comments, I certainly believe your work has merit. I particularly like the growth model curve analyses, which make a lot of sense developmentally. However in my analysis that some major revisions are necessary:

Using zscores and median when comparing children and adults for first ANOVA. I also did not see anything in the results or discussion on the particular point of residual analysis that violated the homogeneity constraint, nor whether you had any power analysis results. Clarifying hand guided movements and representational actions (gestures), removing the rabbit example.

Situating your work compared to Gogate’s, you are not doing Pucini et al.’s (gestures)

Adding new hypotheses in the discussion.

- Thank you for your detailed review of our work! We think your comments improved our work and the manuscript substantially.

Below are in-text my comments:

Introduction

Page 2

Line 12: this sentence seems like an overgeneralization to me. Depending on context, individual differences, samples, the numbers will change. I suggest to add ‘in the context of the studies of XYZ’, ...

- We have now changed this sentence to “In a study by Gogate and colleagues, temporal alignment of parental language and actions in naming events was around

76% for preverbal infants, underscoring the co-occurrence of speech and action in early communication with infants”.

Is this a new method? Is this a modification of a method? Please reference in first paragraph!

- We have now added the preferential looking paradigm as a reference here.

Line 29: we do not know if the pace are similar, generalization here. May be say that they also rapidly learn actions...

- We have now changed the sentence to “Similarly, infants display rapid development in the action domain from early on”.

Lines 30-33: it is one study with star fish shape only (not very ecologically valid) and p at .048. So may be say that a first study using star fish actions suggest that 7 to 9 month olds preferred looking at... than at... , which the authors interpreted as statistical learning.

- We have now removed the sentence and the reference because, as you note, this might reflect weak evidence of infants’ action segmentation and it is not relevant for our argumentation.

Line 35: again, overgeneralization, rephrase.

- We have now changed this sentence to “as mentioned above, Gogate and colleagues found in their study that temporal alignment of actions accompanying caregiver speech was around 76% during year 1”.

Line 54-56: in the study cited, they used representational gestures (symbolic) not movements like Gogate’s. It should be distinguished throughout the paper whether you are talking about gestures or actions or it gets confusing. May be define the terminology you are going to use: actions, movements, gestures...

- We wanted to avoid going into the gestures/actions literature because this would require opening up an entire field which is somewhat tangential to the actions under study. But we have now made it clear in the manuscript whenever we talk about gestures (which is only in the context of the Puccini & Liszkowski study).

Page 3

First paragraph: use ‘may’ rather ‘can’ or results suggest that under certain circumstances, children can...

- We have now changed “can” to “may”.

Do you consider your ‘actions’ Hopping and wiggling ears for rabbits, just as gestures (conventionalized ones) ?

Justification of choice. There is a confusion here for the reader as the experimenter move the tani and löcki whole body = activated movements but the example (hopping and wiggling) provided is about gestures. May be use Gogate examples, that are also about activated movements by a hand.

- We have now clarified here that we refer to manipulation of objects, i.e., parents making the rabbit hop or wiggling its ears rather than a hopping or wiggling gesture performed in the absence of the object.

Line 44 and next: Presenting variability separated from consistency is one way to do it, however literature shows that both are necessary and possibly in interaction, may be suggest this. I believe it is is what you did as you have a within design.

- We totally agree and have now added a paragraph on the importance of both aspects for learning.

Page 4:

Results should not be mixed with hypotheses and so remove line 30-31 and

Add a paragraph on why you selected 18 month olds, and 30 month olds and 3 to 4 year olds based on the literature.

- We chose 18-month-olds and 30-month-olds to capture children on either side of the vocabulary spurt. 3- to 4-year-olds were added on later following the results of the 30-month-olds. We have now rephrased this in the manuscript.

I believe it should be whom not who.

- We have now changed "who" to "whom".

Page 5 methods

Cite reference for preferential looking paradigm test applied to language and intermodal (PLP)

Hirsh-Pasek, & K., Golinkoff, R. M. (1996). The intermodal preferential looking paradigm: onto emerging language comprehension. In D. McDaniel, C. McKee, & H. S. Cairns (Eds.), *Language, speech, and communication. Methods for assessing children's syntax* (pp. 105-124). Cambridge, MA, US: The MIT Press.

It is certainly relevant for 18 months old, reference needed for 3 to 4 year-olds.

- To clarify the methodology, we have cited this reference in the methods section now.

Page 6

How long did the whole procedure last? The training, the testing

- The whole procedure lasted about 4 to 5 minutes and we have added this information in the methods section now.

Experimental design: a graph of the experimental design would help here with the time line, the yoke training, the testing, the attention getter, etc.

- We have now included visual examples for the training and the test phase and hope that these help to clarify the procedure.

Line 52-53. Is this common procedure? Are the % of trials excluded comparable to other studies? Please cite references.

- For young children, data loss around 10% for eyetracking data seems to be frequent in developmental studies. Accordingly, our data loss between 5.9 and 11.4% mirrors data loss in other studies (e.g., Andreu et al., 2013).

Page 7

Line 14 typo

- We have now fixed the formatting issue here.

Design: line 11 please report that it is a between participants design. Anova, etc ... your DV is preferential target looking collapsed over time.

- We have now rephrased this aspect and hope that it is clearer.

Please indicate if you had power measure

- There are few comparable studies looking at children's learning of words with simultaneously presented actions. Studies do exist with regard to the influence of variability on learning but these appear to be underpowered (e.g., Twomey, Ma & Westermann, 2017). We therefore did not have a suitable prior on which to base the power analyses. We therefore decided to test a higher number of children than usual relative to earlier studies, aiming for at least 24 children per condition per age-group.

You are testing consistency and variability independently but children are also exposed to those concomitantly, why did you not consider this, justify.

- We needed to have at least two objects per condition to ensure that we could account for object preferences in our data. Were the design to be a within-subjects design this would have required children to learn 4 novel-word object associations with four different actions presented in the task. We believed this to be too demanding for the age-ranges tested here and decided instead to focus on learning across conditions.

Table 1, what is the unit of measure?

Why is the mean 000, I do not understand these numbers, are they zscores?

- The scores are averaged proportional target looking scores, that have been corrected for participants' looking behavior in the baseline phase. Thus, 0 refers to no changes from the baseline phase. We have now added a sentence to the table caption to facilitate the interpretation of these values: "Scores of 0 reflect that averaged target looking is at chance level, any values above 0 reflect target looking, and values below 0 reflect distractor looking".

Page 8

Growth model curve

Line 53. You start with a theoretical model and then go to 'in practice'. Do you have any references to provide for the 'in practice'? for adults and children? I like the graph, very useful.

- We are very glad you like the graph. We have now added Mirman et al., 2008 as a reference for the adults and Von Holzen et al., 2019 as a reference for the children.

Add a reference for adding a participant id as random factor.

- We have now added Mirman, 2014 as a reference here.

The controls are well thought of and explained.

- Thank you!

As are the linear, quadratic and cubic function of time and also target fixation patterns.

- Thanks again.

Homogeneity violated in residuals.

- Homogeneity was indeed an issue in our analysis, and can often be a problem for looking time data. We transformed our data to log values because this can help to address the issue. However, the problem of homogeneity still remained. We therefore decided to analyse our data without the data transformation but also note that the results have to be interpreted with care.

Line 49 ? power calculations undertaken to ensure that sample sizes were adequate to test the hypothesis considered; are these zscores? They should be, may be use median rather than mean for outliers in zscores.

- There are few comparable studies looking at children's learning of words with simultaneously presented actions. Studies do exist with regard to the influence of variability on learning but these appear to be underpowered (e.g., Twomey, Ma & Westermann, 2017). We therefore did not have a suitable prior on which to base the power analyses. We therefore decided to test a higher number of children than usual relative to earlier studies, aiming for at least 24 children per condition per age-group.

The results are well explained and provide an interesting developmental comparison.

- Thank you! We are glad that the coherences are clear.

It would be important to check the results with your entire data [including the removed data] to see if what you had removed would change the results.

- We have analysed the data again including all outliers, and overall, the results remain the same. Sometimes, p-values are even lower than in the current analysis since mean looking scores that are higher than 2SD above the average are excluded from the reported analysis and may bias the results when not excluded. We are therefore confident that the results are not driven by our exclusion criteria. Indeed, all the data are available on OSF should someone wish to analyse the data in a different way.

Discussion

In Gogate, the setting was quite different a real world situation, with a mom and a 3D object in a meaningful a playful setting. So comparisons may not be granted here, and is another hypothesis as to why you did not find an effect besides the background hypothesis in younger children.

You may want to cite references having found issue with learning from screen, TV, ... in toddlers maybe P. Kuhl: learning from social experience?

- Thank you for suggesting this alternative hypothesis. We have now included it in our paragraph on the 18-month-olds' difficulties to learn words in our setting.

Line 45: Children also learn word independent of action, these are called conventions! Overgeneralization.

- We now explicitly state that this is meant in the context of our study.

Line 46: you cannot use stage with dynamic systems, it is antinomic. You can use developmental time instead (because of individual differences).

- Thank you for highlighting this. We have now changed this sentence to "In terms of a dynamic systems account, this development reflects how the interaction between

the learner and the complexity of the multimodal environment plays a role in shaping the learning experience”.